# High flavivirus structural plasticity demonstrated by a non-spherical morphological variant

Seamus R. Morrone[1,2,6], Valerie S. Y. Chew[1,2,6], Xin-Ni Lim [1,2,6], Thiam-Seng Ng[1,2], Victor A. Kostyuchenko[1,2], Shuijun Zhang[1,2], Melissa Wirawan[1,2], Pau-Ling Chew[1,2], Jaime Lee[1,2], Joanne L. Tan[1,2], Jiaqi Wang[1,2], Ter Yong Tan[1,2], Jian Shi[2,3], Gavin Screaton[4], Marc C. Morais[5] & Shee-Mei Lok [1,2 ✉]

Previous flavivirus (dengue and Zika viruses) studies showed largely spherical particles either with smooth or bumpy surfaces. Here, we demonstrate flavivirus particles have high structural plasticity by the induction of a non-spherical morphology at elevated temperatures: the club-shaped particle (clubSP), which contains a cylindrical tail and a disc-like head. Complex formation of DENV and ZIKV with Fab C10 stabilize the viruses allowing cryoEM structural determination to ~10 Å resolution. The caterpillar-shaped (catSP) Fab C10:ZIKV complex shows Fabs locking the E protein raft structure containing three E dimers. However, compared to the original spherical structure, the rafts have rotated relative to each other. The helical tail structure of Fab C10:DENV3 clubSP showed although the Fab locked an E protein dimer, the dimers have shifted laterally. Morphological diversity, including clubSP and the previously identified bumpy and smooth-surfaced spherical particles, may help flavivirus survival and immune evasion.

[1] Program in Emerging Infectious Diseases, Duke-NUS Medical School, KTP Building, 8 College Road, Singapore 169857, Singapore. [2] Centre for BioImaging Sciences, National University of Singapore, Singapore 117557, Singapore. [3] CryoEM Unit, Department of Biological Sciences, National University of Singapore, Singapore 117557, Singapore. [4] Medical Sciences Division, University of Oxford, Level 3, John Radcliffe Hospital, Oxford OX3 9DU, UK. [5] Sealy Center for Structural Biology and Molecular Biophysics, Department of Biochemistry and Molecular Biology, University of Texas Medical Branch, Galveston, TX 77555, USA. [6] These authors contributed equally: Seamus R. Morrone, Valerie S. Y. Chew, Xin-Ni Lim. ✉email: sheemei.lok@duke-nus.edu.sg

Zika (ZIKV) and dengue viruses (DENV) are members of the *Flaviviridae* family, which also includes other important human pathogens, such as West Nile and yellow fever viruses. DENV consists of four main serotypes (DENV1, DENV2, DENV3, and DENV4)[1]. Infection with one serotype confers lifelong protection against that particular serotype. However, in a secondary infection with a different serotype, the antibodies elicited from the primary infection may not only be unable to neutralize the second serotype, but instead the antibody bound to the virus may enhance infection of the Fcγ receptor-positive myeloid cells including macrophages, leading to increased disease severity[2]. This normally occurs at sub-neutralizing antibody concentrations. This complicates the development of vaccines, which would need to induce robust protective responses against all four serotypes simultaneously. In addition to the presence of four serotypes, some studies also showed that DENV particles are present in different morphologies[3–5], further complicating vaccine development. ZIKV has a similar structure as DENV but has a heightened structural stability[6] and can cause debilitating anomalies in fetuses of infected pregnant women[7].

The core of the flavivirus particle, consisting of the positive-sense RNA genome complexed with capsid proteins, is surrounded by a lipid bilayer membrane. On its surface are 180 copies of the icosahedrally-arranged lipid membrane-anchored envelope (E) and membrane (M) protein heterodimer[8,9]. The E protein is the major surface protein and consists of three ecto-domains: DI, DII, and DIII[10] (Supplementary Fig. 1a). The E protein facilitates entry of virus into host cells, i.e., attachment to cell receptor(s), as well as fusion with the endosomal membrane to release the viral genome into the cell. DIII of E protein is thought to be a component of receptor binding, and the tip of DII contains a fusion loop that directly interacts with the endosomal membrane during the fusion process[10].

Some strains of mature DENV2 can display two different types of morphologies[3,4] when temperature is switched from 28 to 37 °C: from the compact smooth to the bumpy-surfaced particles (Supplementary Fig. 1a,b). This indicates that the morphology of these DENV2 strains in mosquito (physiological temperature 28 °C) is different from that in humans (37 °C). Other DENV2 strains are more stable, maintaining the compact smooth surfaces[11]. On the compact-surfaced mature virus, the E proteins have extensive tight interactions with each other, resulting in three sets of dimers lying parallel to each other forming a raft (Supplementary Fig. 1a). Thirty such rafts are then arranged in a herringbone pattern on the virus surface[8,9]. Bumpy-surfaced mature virus particles have a larger radius than smooth particles, as a result of looser interactions between E proteins on the particle surface (Supplementary Fig. 1b). While most of the E proteins remain as dimers, protomers within a dimer at the icosahedral twofold vertices separate from each other[3]. Highly neutralizing human antibodies have been shown to bind across E proteins, and thus are dependent on the quaternary structure of the smooth compact virus structure[12,13]; since the E protein arrangement is very different on the bumpy-surfaced virus particles, these antibodies may not be able to neutralize them. This suggests that the morphological diversity of DENV could allow the virus to evade the human immune system. In addition, antibodies that enhance virus infection are identified to be directed against the fusion loop of the E protein[14]. On the compact mature virus surface, the fusion loop is partially hidden between DI–DIII of the opposite E protein protomer within the dimer conformation (Supplementary Fig. 1a). On the other hand, on the bumpy-surfaced virus structure, since the E protein arrangement is looser, the fusion loop is more accessible and thus could likely elicit more anti-fusion loop antibodies. Therefore, understanding all possible morphological variants of flavivirus is important for the design of both effective anti-viral therapeutics and vaccines.

In this paper, we show a flavivirus morphological variant, the club-shaped structure, which has not previously been reported. We show it is a temperature-dependent induced structure, and we observe this morphology in multiple DENV strains and ZIKV, indicating that this is common amongst flaviviruses. We solve the cryoEM helical structures of an antibody-virus complex for both DENV and ZIKV in this morphology and reveal the architecture of these structures. We also demonstrate these variants retain wild-type attachment properties as well as unique antibody recognition. All these have important implications for flavivirus biology and for therapeutic and vaccine design.

## Results

**DENV exhibits temperature-dependent structural changes.** Most of the previous structural work has been performed with DENV2; here, we show a non-spherical morphological variant, the club-shaped particles (clubSPs) of DENV3-CH53489, which accounts for about half of its virus population (Fig. 1a and Supplementary Fig. 2a) when the purified virus sample is incubated at 29 or 37 °C for 30 min. Of those particles that remained spherical at 29 °C, most were smooth-surfaced, while those at 37 °C were bumpy (Fig. 1a). We also tried different incubation time (15 min to 2 h) at 37 °C to determine if more or less clubSP will form. Results showed that the fraction of particles turning to clubSP are the same regardless of incubation time (Supplementary Fig. 2c). To determine the temperature that triggers the formation of clubSPs, the purified DENV3-CH53489 that was grown in mosquito cells was incubated at a series of temperatures ranging from 4 to 40 °C, and the percentage of the clubSPs was calculated. We observed a gradual increase in percentage of clubSP virus population, the maximum reached at 37 °C (Fig. 1b). Testing of clubSP formation of the mammalian Huh7 cells-derived DENV3-CH53489 at 4, 29, 37, and 40 °C also showed marked increases in clubSP at temperatures ≥29 °C (Supplementary Fig. 3). Because the samples had been stored on ice prior to freezing on cryo-electron microscopy (cryoEM) grids, this suggests that the formation of clubSPs from the original spherical particles is a temperature-irreversible structural change. Close examination of micrographs of other viruses (DENV1, DENV2 and ZIKV) at 37 °C also showed the presence of some clubSPs (Supplementary Fig. 2b). Incubation of DENV1-Westpac and DENV2-PVP94/07 at 4, 29, 37, and 40 °C showed a sharp increase in the formation of clubSPs at temperatures ≥37 °C (Supplementary Fig. 3). This indicates clubSP formation is neither a product of the specific cell type used to produce the virus, nor a strain-specific property.

**Antibodies recognize the flavivirus structural variant.** To ascertain that the clubSPs are indeed flavivirus particles, we incubated the DENV3-CH53489 and ZIKV samples with antibodies specific to E proteins before and after incubation at 37 °C. One of the antibodies is a neutralizing mouse monoclonal antibody 8A1[15], we have determined its PRNT_{50} against DENV3-CH53489 to be 0.11 μg/ml. When the antigen-binding fragment (Fab) of the DENV3 anti-E protein DIII antibody 8A1[15] was added to DENV3-CH53489 before it was exposed to 37 °C, we observed only spiky spherical particles (Fig. 1c, left) instead of a large percentage of clubSPs as observed in the 37 °C uncomplexed control (Fig. 1a, iii). This suggests that the Fab inhibits the ability of the virus to undergo the structural changes necessary to form clubSPs at 37 °C. When Fab 8A1 was added to the virus sample that was pre-incubated at 37 °C, the Fab bound to the head and the tip of the tail of the preformed clubSPs, as indicated by the increased spikiness of these parts of the particles (Fig. 1c, right)

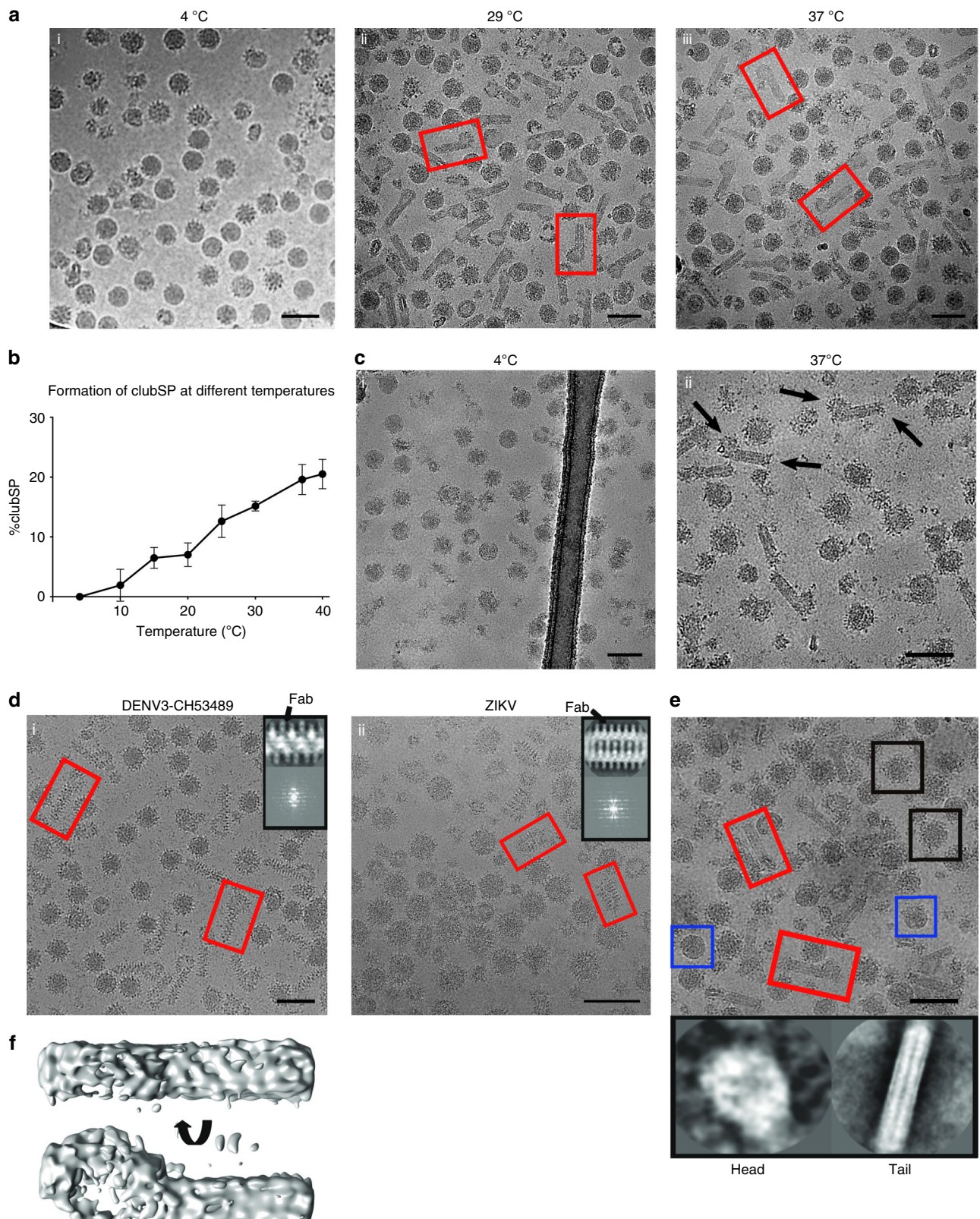

compared with control (Fig. 1a, iii). This suggests that DIII of the E protein is more exposed in these regions, allowing Fab 8A1 to bind. We tested the exposure of the fusion loop of E protein on the preformed DENV3-CH53489 clubSPs by incubating them with the Fab fragment of the fusion-loop antibody 4G2. CryoEM micrographs showed the head of the clubSPs appeared spikier

(Supplementary Fig. 4a, center) than the smooth-surfaced uncomplexed control (Supplementary Fig. 4a, left), suggesting that the E protein fusion loop on the head of the clubSPs is accessible. We also tested the neutralization profile of antibody 4G2 on DENV3-CH53489 clubSPs (which were induced at either 29 or 37 °C), and showed that 4G2 is unable to neutralize these

**Fig. 1 DENV3-CH53489 particles induced by temperature to form clubSPs. a** Micrographs of DENV3-CH53489 incubated at various temperatures. A large proportion of the particles transform from spherical to clubSPs (a representative is boxed in red) at 29 and 37 °C. Micrographs represent one of eight independent experiments. **b** Graph showing percentage of clubSPs increase gradually with temperature. The maximum percentage of clubSP is reached at 37 °C. Data are presented as mean values with error bars represent standard deviations (±SD) calculated from three individual experiments. Source data are provided in the source data file. **c** Representative micrographs of anti-DIII E protein Fab 8A1 added to DENV3-CH53489 before (i) or after (ii) virus is exposed to 37 °C. In (i), Fab 8A1 inhibits clubSP formation, whereas (i) there are similar amounts of clubSP, as the uncomplexed control in (**a**). (Right) The Fab binds to the head (open arrow) and also the tip of the tail (closed arrow) of DENV3 clubSPs. $n = 3$. **d** When Fab C10 is added to DENV3-CH53489 (i) and ZIKV (ii) prior to exposure to 37 °C, the surface becomes spiky, indicating Fab binding. Upper right corner on each micrograph shows 2D average (top panel) of the projections, which are boxed sequentially along the filaments with partially overlapping regions, and also their power-spectrum (bottom panel) of the central sections of the tail portion of clubSP or the body of catSP. Fab density is indicated by an arrow in the 2D averages. A representative clubSP or catSP is boxed in red in the micrographs. Micrographs represent data from two independent experiments for both DENV3-CH53489 and ZIKV. **e** Micrograph showing the DENV3-CH53489 clubSPs (red box) are mostly mature virus, as anti-prM Fab DV62.5 binding was not detected in the 2D averages (bottom panels) of either the head or tail of the clubSP particle. Fab DV62.5 was observed to bind to the spherical spiky immature virus population (black box) as shown in Supplementary Fig. 4b. Uncomplexed smooth-surfaced mature spherical virus is boxed in blue. Scale bars in micrographs represent 100 nm. Micrographs are from one experiment. **f** CryoEM map of asymmetric reconstruction of the DENV3-CH53489 clubSPs shown in different views.

viruses at all concentrations tested (Supplementary Fig. 4a, right). This shows that the clubSP particles can have different levels of exposure of certain epitopes on various parts of the structure, and therefore some antibodies likely could not achieve full occupancies. This may help the virus to escape from antibody neutralization.

Antibody C10[13,16], a flavivirus cross reactive E protein dimer binding antibody, is able to neutralize DENV3-CH53489 clubSP structures that were induced either at 29 °C ($PRNT_{50} = 3.44\ \mu g/ml$) or 37 °C ($PRNT_{50} = 2.58\ \mu g/ml$). Antibody C10 had also been shown previously to be highly neutralizing against ZIKV ($PRNT_{50} = 0.024\ \mu g/ml$)[13]. We next observed the binding of Fab C10 to both DENV3-CH53489 and ZIKV by using cryoEM (Fig. 1d, left and right, respectively). When we added Fab C10 to DENV3-CH53489 and then increased the temperature to 37 °C, half of the virus particles become clubSPs, similar to the uncomplexed controls, except Fab was observed to bind to the entire surface of the clubSPs (Fig. 1d, left). The same is also observed when the virus is incubated first at 37 °C to induce clubSP and then Fab C10 is added (Supplementary Fig. 4c). Interestingly for ZIKV, the Fab complexed virus particles (Fig. 1d, right) have a caterpillar appearance (termed catSP) different from the clubSP of its uncomplexed control (Supplementary Fig. 2b). Therefore, this suggests that this morphological difference could be a function of how the antibody interacts with ZIKV.

To determine if the clubSPs originate from particles that are highly immature, the Fab of an anti-prM antibody, DV62.5, was added to the sample. DENV3-CH53489 clubSPs have either low or no Fab bound to either the tail or head portions (Fig. 1e), as shown by 2D averages (Fig. 1e, bottom left and right, respectively). The 2D averaged spiky spherical immature virus in the same sample showed a larger radius of spikes with extra densities corresponding to Fabs, compared with the uncomplexed immature virus (Supplementary Fig. 4b). These results suggest that the DENV3-CH53489 clubSPs are likely not formed from highly immature virus particles. We also prepared immature DENV3-CH53489 by growing virus in the presence of ammonium chloride. We obtained in the DENV3-CH53489 population ~60% particles that were highly immature (spiky surfaced), ~7% fully mature (smooth surfaced) and ~30% partially immature (half spiky and half smooth surfaced) (Supplementary Fig. 5). Incubation at 29 °C showed an increased percentage of clubSP in the virus population from 0.1 to 23%. While the percentage of highly immature particles remained largely the same, that of the mature and partially immature virus were reduced (Supplementary Fig. 5). This suggests that the clubSP does not originate from the highly immature particles.

**ClubSPs contain RNA and can attach to cells.** The DENV3-CH53489 spherical particles at 4 °C showed they are mostly full particles with RNA packaged inside, as observed by their dark centers (Fig. 1a, left), compared with some DENV2 strains where, although rarely observed, some particles appear to have partially filled or empty cores (whiter center) (Supplementary Fig. 2d). It is possible that the formation of DENV3-CH53489 clubSPs is due to the loss of its RNA genome. We detected for the release of RNA outside the viral particle after induction of clubSP by their sensitivity toward added RNase—viral genome RNA inside particles are protected from RNase while those released are not. We first incubated the virus at different temperatures and then conducted RNase digestion. RNA digestion is stopped before lysing the virus particle and then qRT-PCR is conducted using a set of primers to detect for a stretch of the undigested DENV3-CH53489 RNA. The results showed that there was no decrease in the amount of viral genome after temperature-induced structural changes of DENV3-CH53489, suggesting that the genome is not extruded from the particle (Fig. 2a, left). We also conducted a control experiment to show that when virus is lysed first (thereby releasing its viral RNA genome), it is sensitive to RNase digestion, and qRT-PCR using the same set of primers showed a dramatic reduction of detectable amounts of viral genome compared with the sample where no RNase was added (Fig. 2a, right), indicating that these primers are sufficient to detect viral RNA digestion.

To test how formation of clubSP in virus samples affects the attachment of virus to mosquito (C6/36) and mammalian (BHK21) cells, we first pre-incubated the purified DENV3-CH53489 (cryoEM quality) at 4, 29 or 37 °C for 30 min. ClubSP formation will be induced in the 29 and 37 °C treated samples. All samples were then cooled to 4 °C before adding to pre-cooled cells, thus ensuring the virus will not be endocytosed. Unbound virus was washed away at 4 °C, and then the amount of virus on the cell surface was quantified by qRT-PCR. Results showed no difference in the amount of virus attached for all temperature-pretreated virus samples to both C6/36 and BHK cells at 4 °C (Fig. 2b). This suggests that formation of clubSP in the 29 and 37 °C treated virus sample did not alter the ability of virus to attach to either C6/36 or BHK cells.

**Structural reconstruction of clubSP and catSP.** To determine the uncomplexed whole DENV3-CH53489 clubSP structure, we conducted asymmetric single-particle reconstruction. However, only a low-resolution map was obtained showing a disc-like head and a tubular tail (Fig. 1f). While the overall features were apparent and consistent with the 2D class-average

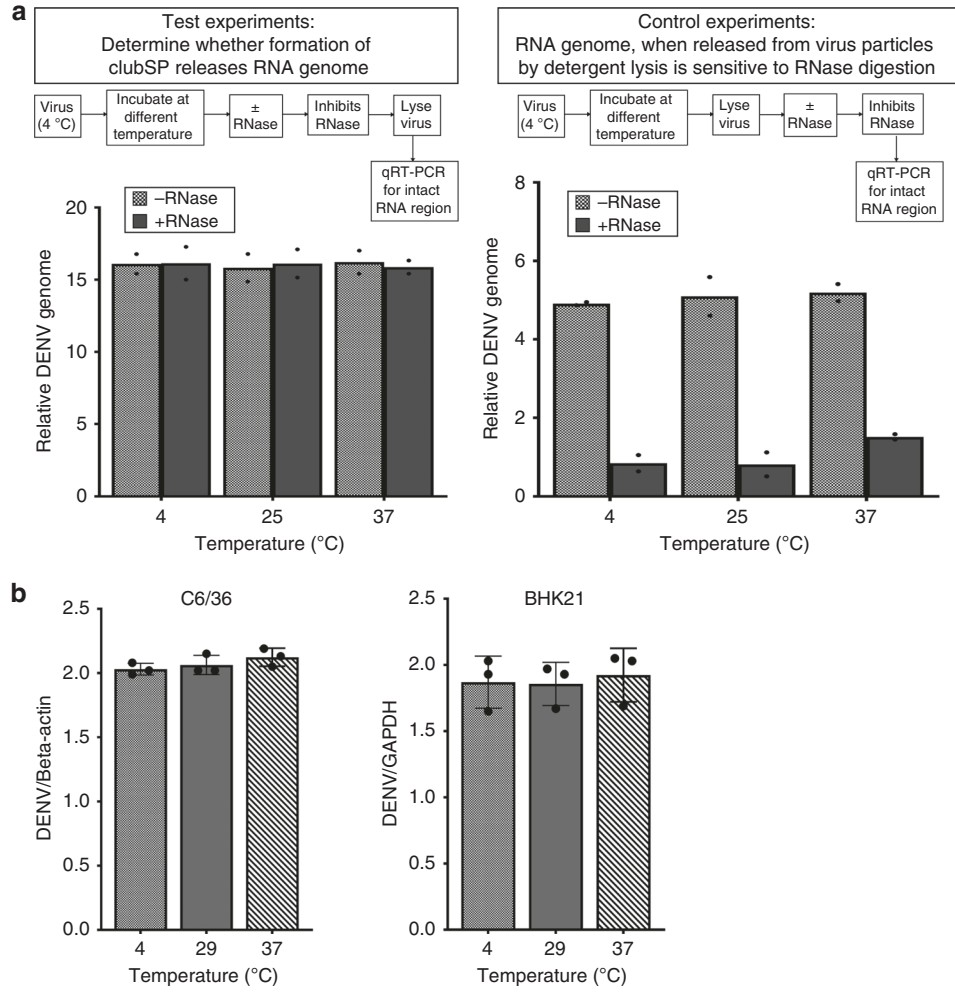

**Fig. 2 DENV3-CH53489 clubSPs contain RNA and remain infectious. a** The viral DENV3-CH53489 RNA genome is not extruded out of the particle after temperature-induced structural change at 25 and 37 °C. (Left panel) Viruses at 4 °C were transferred to different temperatures and incubated for 30 min. RNase is then added to digest away any RNA genome that was extruded outside the virus. RNase activity is inhibited and then the virus was lysed. Detection of a 260-nucleotide stretch of RNA genome using a primer set in qRT-PCR indicates intact viral genome. In all temperatures, the amount of genome with and without addition of RNase appeared the same, suggesting that viral RNA remains within the particle after temperature treatment and were thus protected from RNase treatment. The y-axis is expressed as difference in Cq values from virus-free reference. (Right panel) Control experiments to show when virus is lysed first before exposure to RNase, the RNases can successfully break down viral RNA genome. Results showed that when RNA genomes (due to lysis) are exposed, RNases digestion occurs and there are much lower amounts of intact RNA genome detected by qRT-PCR than when no RNase is added. **b** Attachment assay showing 29 and 37 °C pretreated viruses containing clubSPs have the same binding capacity to both C6/36 and BHK21 cells as the 4 °C sample. Data are presented as mean and error bars represent SD calculated from three individual experiments; each experiment was done in duplicate. The y-axis shows the relative DENV3-CH53489 RNA against housekeeping genes: beta-actin and GAPDH for C6/36 and BHK21 cells, respectively. Source data are provided in the source data file.

(Supplementary Fig. 2a), details of the E protein organization were non-existent, likely due to the high conformational heterogeneity of the whole viral particle. Examination of the sizes of the head and also the length of the tail in individual particles confirmed the presence of heterogeneity. However, the radii of many of the tails are consistent to each other, indicating that central segments of the tail structure could possibly be aligned and averaged for higher resolution 3D image reconstruction. We thus attempted to solve the structure of this part of the clubSP tail. The middle sections of the tails were boxed and their 2D averages showed a smooth-surfaced structure with a width of ~200 Å (Supplementary Fig. 6a, left). Fourier transform of the 2D average showed only one major layer line, suggesting that it is either non-helical or disordered or cannot be aligned to each other due to the smooth surface (Supplementary Fig. 6a, right). In contrast, the Fab C10 bound clubSP DENV3-CH53489 tail and

the catSP ZIKV showed clear, distinct features in the 2D averages that correspond to the Fab molecules (Fig. 1d, top panel in upper right corner of the micrographs). Fourier transforms of these 2D averages of both types of particles showed clear layer lines (Fig. 1d, bottom panel in upper right corners of micrographs), indicating the presence of helical symmetry. This could be due to Fab C10 stabilizing the structures and also providing additional features for more accurate alignment. However, the indexing of the layer lines did not permit unambiguous assignment of initial helical parameters, possibly due to spots in the layer lines blurring/merging together. We therefore performed asymmetric single-particle reconstructions of the boxed segments of both the Fab C10:DENV3-CH53489 clubSP tail and the whole Fab C10:ZIKV catSP (Supplementary Fig. 6b,c). The resulting density allowed for fitting of the previously solved Fab C10:ZIKV recombinant E protein dimer cryoEM model[13] (PDB: 5H37). The

rise and rotation required to superimpose one E protein dimer asymmetric unit (asu) onto the adjacent asu provided the initial estimated parameters for helical reconstruction. These values were then refined to obtain the final helically reconstructed map. Both the Fab C10:ZIKV catSP and the Fab C10:DENV3 clubSP tail have C1 point group symmetry. However, they have different helical parameters: the Fab C10:ZIKV catSP helical parameters consist of a 24.1° rotation and 8.6 Å axial rise per subunit, whereas the Fab C10:DENV3 clubSP tail helical parameters were a 102.3° rotation and 18.5 Å axial rise. These differences in helical parameters lead to dramatically different structures. The cryoEM map of the Fab C10:ZIKV catSP and Fab C10:DENV3 clubSP viruses were determined to 9.4 Å and 10.4 Å resolution, respectively, as measured using the gold standard Fourier-Shell Correlation (FSC) (Supplementary Fig. 7a, c, Supplementary Table 1). The structures of the Fab C10 complexed with whole spherical ZIKV[13] and also the recombinant DENV2 E protein[16] had been solved previously (Supplementary Fig. 1c, d). These complexes were fitted as rigid bodies into their corresponding densities in the final clubSP and catSP helical maps with no clashes between the fitted molecules, and all densities corresponding to the E protein ectodomain layer were accounted for (Supplementary Fig. 7b, d).

The Fab C10:ZIKV catSP helical portion has a 270 Å outer diameter and 130 Å inner diameter, excluding the densities corresponding to the Fabs (Fig. 3a, middle panel). The resolution of the density map allowed us to fit the entire E and M proteins, including their transmembrane regions, and the variable regions of Fab C10 (Supplementary Fig. 7b). The asymmetric unit contains two Fab C10 molecules each binding across either end of the E protein dimer; by applying the helical parameters, a filament can be generated of the central portion of the catSP (Fig. 3b). Three neighboring asymmetric units within the helical structure are similar in conformation to a raft of the previously published Fab C10 complexed with the smooth spherical ZIKV particle[13]; the superposition of the three fitted dimers and the raft from the spherical ZIKV particles (Fig. 3c) shows a root-mean-square deviation (RMSD) value of 0.8 Å. The previous near-atomic resolution structure of Fab C10:spherical ZIKV[13] showed Fab C10 binding across E proteins at the intra-dimer interface and also across E proteins at the inter-dimer interface, thus locking individual protomers within the whole E protein raft together as a structural unit (Supplementary Fig. 1d). Thus, the formation of the catSP structure may be a function of the antibody locking the raft as a rigid structural unit, leading to a morphology different to the clubSP of the uncomplexed ZIKV structure (Supplementary Fig. 2b), where individual E protein dimers are free to separate.

The cryoEM map of the central segments of the helical tail of Fab C10:DENV3-CH53489 clubSP has a 150 Å outer diameter and a 30 Å inner diameter (Fig. 3d, middle). The densities corresponding to the ectodomain of the E proteins and the variable region of the Fab C10 are well resolved, whereas those corresponding to the transmembrane regions of E and M proteins are much poorer, therefore only the E protein ectodomains complexed with Fabs were fitted. An asymmetric unit consists of an E protein dimer complexed with two Fab C10 molecules. The E protein dimers (Fig. 3e) are organized very differently from their arrangement in the Fab C10:ZIKV catSPs structure (Fig. 3b). Potential inter-E dimer interactions in both the catSP and clubSP structures were identified by measuring an 8 Å cutoff distance between pairs of Cα atoms; fewer interactions were identified in the clubSP compared with catSP structure (Fig. 3f). The previously solved crystal structure of Fab C10 complexed with soluble DENV2 E protein shows the Fab binding across the E protein protomers within a dimer, thus locking them together[17].

The cryoEM Fab C10:spherical ZIKV structure shows additional C10 binding to E proteins across the E protein inter-dimer interface, thus locking the entire raft[13]. In our cryoEM structure of the helical tail of the DENV3 clubSP, the antibody binds across the E protein protomers within a dimer, but not across the E inter-dimer interface as observed in ZIKV[13], therefore allowing movements of a dimer relative to its neighbor. Alignment of our clubSP E protein ectodomain dimer with that of the crystal structure of Fab C10:DENV2[17] (PDB: 4UT9) and also that of the uncomplexed spherical DENV3 structure[18] (PDB: 3J6T), showed that the curvature of the E protein in the clubSP structure is more similar to the flatter curvature of the crystal structure (Supplementary Fig. 8). The curvature of the E proteins on spherical virus particles is maintained by the interaction of the underside of the E ectodomain with the membrane-associated stem regions of both the E and M proteins; since the curvature of the E ectodomain of our clubSP structure is flatter, this suggests that it may have dissociated from these interactions.

## Discussion

Examination of the Fab C10:DENV3-CH53489 clubSP tail shows that it likely cannot accommodate the RNA genome in this part of the structure, based on its inner diameter. Measurement of the head of the 2D average of the DENV3 clubSP (Supplementary Fig. 2a) and calculation of its volume treating it as a cylinder suggests that there is room to accommodate about two copies of viral genome (Supplementary Table 2). The cylindrical shape of the clubSP head was observed in our asymmetric reconstructed map (Fig. 1f). As for the Fab C10:ZIKV catSP, the calculation of the inner volume of its body based on its asymmetric reconstruction suggests that it can accommodate at least two viral genomes, whereas in the spherical particles, ~4 copies of genome (Supplementary Table 2) are possible. Ordered density corresponding to the RNA genome has not been observed in any single-particle reconstruction of flavivirus of any morphology[3,6,12] likely because the genome is structured differently in individual particles and are therefore cannot be aligned for averaging during the reconstruction process.

Based on the structures reported here and previous DENV/ZIKV-Fab complexes, we propose structural rearrangements of E proteins cause the change from the spherical particles to either the catSP or the clubSP. In the Fab C10:ZIKV spherical and catSP structures, all E proteins within a raft are locked by the antibody[13], thus rafts move as rigid structural units, and only relative motions between entire rafts can occur (Fig. 4a). Higher temperatures presumably provide enough energy to disrupt the inter-raft interface, leading to a rotation of the raft, and subsequent relaxation into a configuration utilizing interactions similar to the inter-dimer E protein interface. Thus the same strong inter-dimer interaction that occurs within the raft, also drives formation of the helical structure to form the catSP. As for the Fab C10:DENV3-CH53489 clubSP tail structure, the Fab C10 locks only the E protein protomers within a dimer, and thus dimers within a raft can segregate. In the transition from the original spherical structure to the final helical structure, the E protein dimers need to move with respect to each other (Fig. 4b) in addition to the relative movements between adjacent rafts. This suggests that the E protein inter-dimer and inter-raft interactions are both weaker in DENV3. For both clubSP and catSP, the attachment of the E protein dimers to their underlying membrane via their transmembrane regions presumably keep E proteins from diffusing away from the particle while transitioning between different structures. Our inability to reconstruct the tail of the uncomplexed DENV3 clubSP structure, even when we imposed similar helical parameters as the Fab C10:DENV3 clubSPs, suggests that

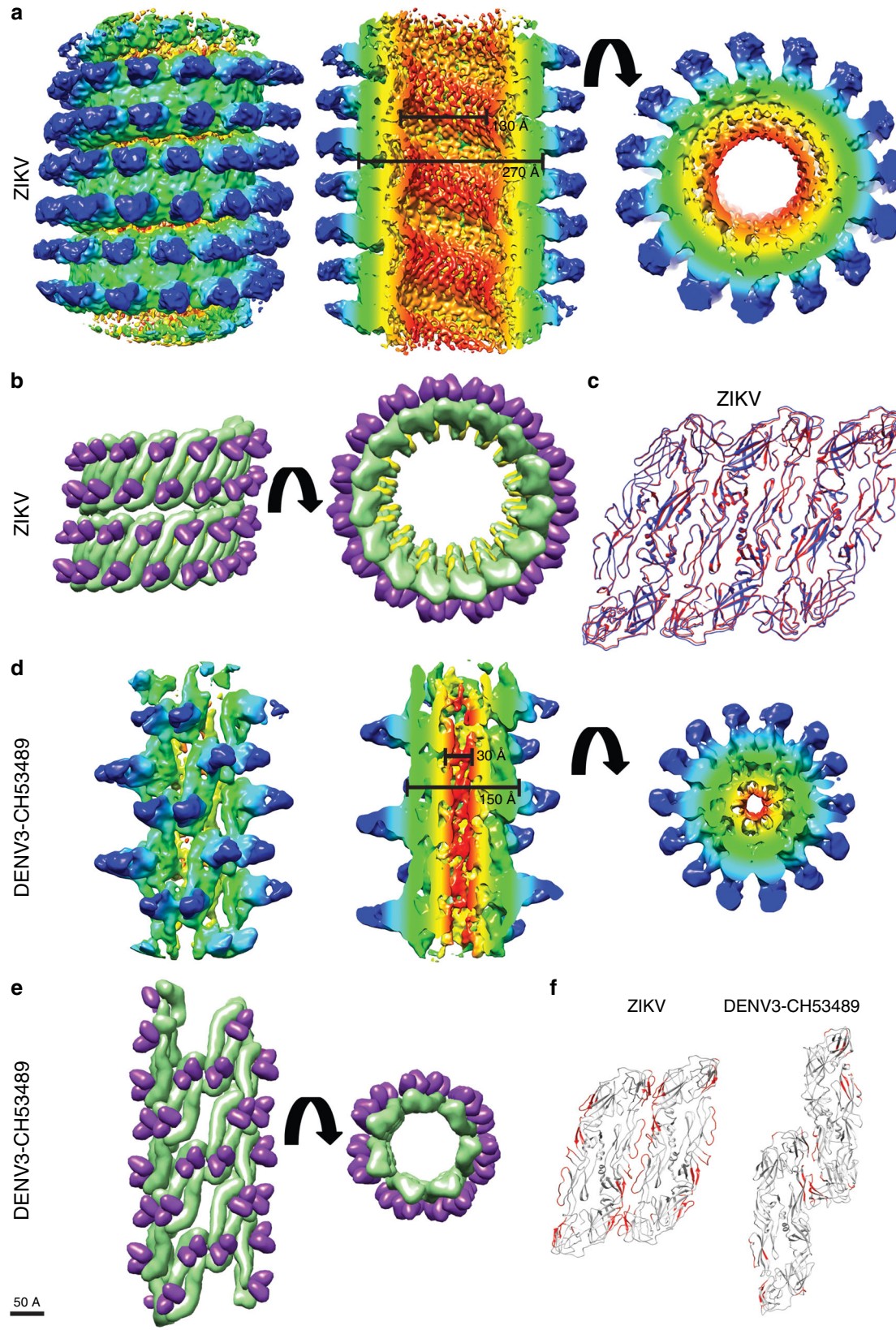

the E protein organization is even less ordered than that of the Fab C10:DENV3 clubSPs. Results presented by Rey and colleagues[17] showed that when C10 is added to mostly monomeric soluble E proteins, the C10 Fabs can assemble the E proteins into dimers. Whether the E protein protomer interactions within a dimer in the uncomplexed DENV3:clubSP are also weak is hard

to determine, but the protomers should be near each other so that Fab C10 can assemble the dimer. Other than C10, there is also another reported quaternary structure dependent binding human monoclonal antibody 5J7[12], which binds across three E proteins on the spherical compact virus surface (Supplementary Fig. 1e). This epitope would be completely disrupted in the tail part of the

**Fig. 3 Helical structures of Fab C10:ZIKV catSP and the tail of Fab C10:DENV3 clubSP. a–c** The cryoEM helical structure of Fab C10:ZIKV catSP. **a** The cryoEM map surface (left) and different slice-through views (middle and right panels). The map is colored by the regions containing (1) the inner leaflet of the bilayer lipid membrane (red), (2) the transmembrane regions of the E and M proteins (yellow), (3) the ectodomain of the E and M proteins (green) and (4) variable (cyan) and constant regions (blue) of the Fab C10. It also correspond to cylindrical radius: red (30-50 Å), yellow (51-100 Å), green (101–125 Å), cyan (126-150 Å), blue (151-175 Å). **b** Two turns of the fitted helical structure in two different views. The E proteins, M proteins and the variable region of the Fabs are colored in green, yellow, and purple, respectively. **c** The superposition of three adjacent asymmetric units consisting in total three E protein dimers of the catSP helical structure (red) with an E protein raft (blue) from the icosahedral spherical mature particle indicating the three asymmetric unit is nearly structurally identical to the raft structure (RMSD is 0.8 Å). **d, e** The helical cryoEM structure of Fab C10:DENV3 clubSP. **d** The surface (left) and different slice-through views (middle and right panels) of a segment of the tail of Fab C10:DENV3 clubSP cryoEM map. The map is also colored according to their regions containing different parts of the surface proteins and Fabs similar to (**a**), but their cylindrical radius are as follows: red (5–10 Å), yellow (11–30 Å), green (31–60 Å), cyan (61–90 Å), blue (91–120 Å). **e** Approximately two turns of the fitted homology model of Fab C10:DENV3-CH53489 clubSP helical structure in different views, colored as in (**b**). **f** Inter-dimer contacts between E protein dimers of catSP (left) and clubSP (right) are very different. There are far fewer E protein inter-dimer contacts in the Fab C10:DENV3 clubSP than the Fab C10:ZIKV catSP. The contacts are colored in red and were identified by using a cutoff distance of 8.0 Å between Cα residues.

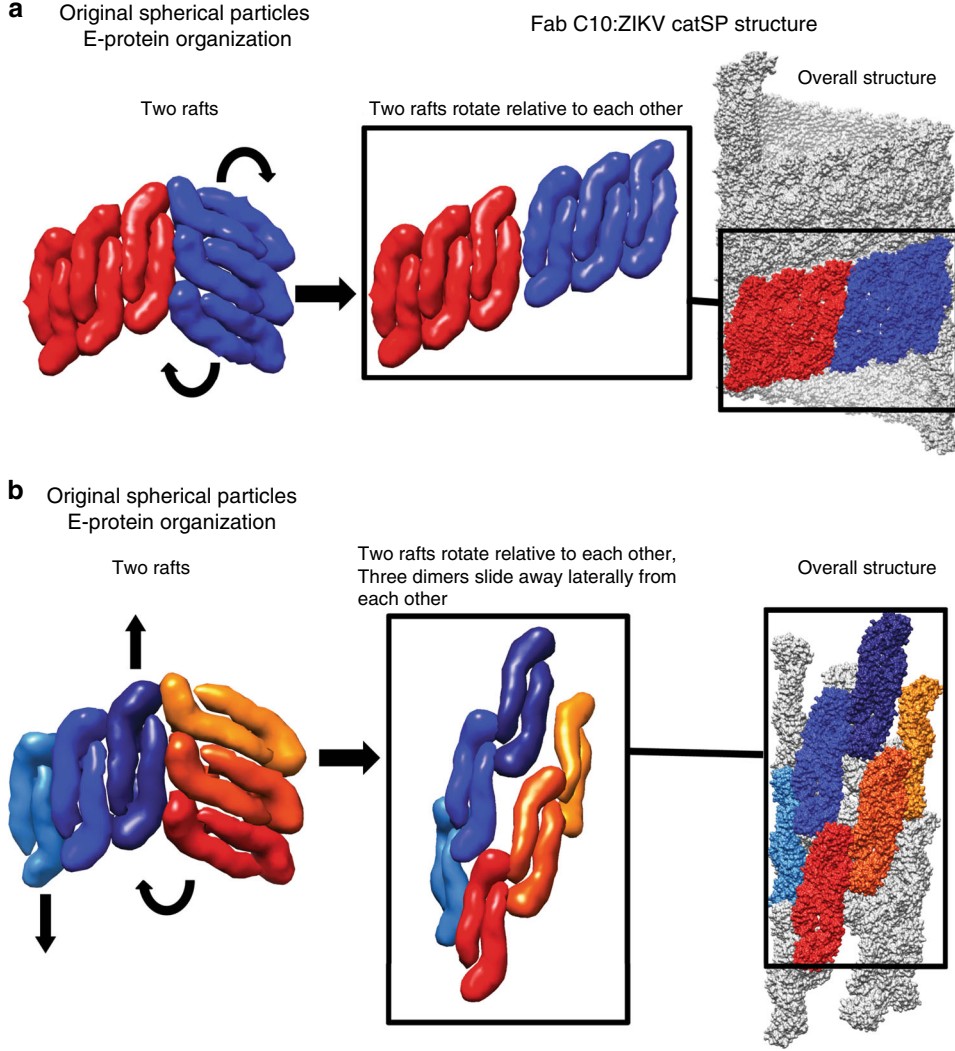

**Fig. 4 E proteins rearrangement on the original spherical particles to form catSP and clubSP. a** The Fab C10:ZIKV catSP structure shows when the E protein raft is locked by the Fabs as observed previously by the near-atomic resolution cryoEM structure of Fab C10:ZIKV spherical structure[13], the rafts of the original spherical ZIKV particle (left) at elevated temperatures have to rotate relative to each other (curved arrows in left panel) to generate catSP helical structure (middle and right panels). Two rafts are shown in red and blue. **b** The Fab C10:DENV3 clubSP structure shows that Fab C10 does not lock the E dimers within the raft in DENV3, as the dimers have moved relative to each other. This suggests that, when incubating the spherical particles at elevated temperatures, structural rearrangement may include rotation of the rafts relative to each other (as observed in ZIKV catSP, curved arrow in left panel) and also the lateral motions (indicated by vertical arrows in left panel) of the E protein dimers, achieving the final helical structure (middle and right panels). These suggest that the inter-dimer interactions between the E proteins are also weak in addition to their inter-raft interactions. Two rafts each with three dimers are colored as different shades of red or blue and displayed as surface representations.

DENV3 clubSP suggesting that 5J7 is likely unable to bind. An enhancing mouse monoclonal antibody 2A10G6 binds to the fusion loop of the E protein (Supplementary Fig. 1e). As the E proteins are more loosely packed in the tail part of clubSP, it may be able to bind more efficiently.

ClubSP is another alternative structure of the flavivirus in addition to the smooth and bumpy surface particles, and this structural polymorphism may confer advantages in the survival of the virus. In addition, the level of exposure of certain epitopes differ in different parts of the clubSP (head or tail), which may help the virus escape from the immune system.

## Methods

**Antibodies and viruses**. Monoclonal antibodies 8A1, DV62.5, and C10 were obtained from Aravinda de Silva, Davide Corti, and Gavin Screaton, respectively. Antibody 4G2 was purchased from ATCC. The virus strains used were as follows: DENV1-WestPac, DENV1-PVP159, DENV2-NGC, DENV2-PVP94/07, DENV3-CH53489, ZIKV-H/PF/2013.

**Preparation of Fabs**. For antigen-binding fragment generation, the IgG of the respective antibody (5–8 mg/mL) was incubated 2–6 h (optimized for each IgG) at 37 °C with papain conjugated resins (Thermo Scientific). The resins were removed by centrifugation and the Fab fragment was then further purified by anion-exchange (Resource Q, GE Healthcare) and gel filtration (Superdex200 increase 10/300 GL, GE Healthcare) on an AKTA purifier system (GE Healthcare).

**Virus sample preparation**. *Aedes albopictus* C6/36 mosquito cells (ATCC) were infected with virus at an MOI = 1, incubated for 2 h at 29 °C, and the inoculum removed and replaced with fresh RPMI-1640 medium and 2% fetal bovine. Virus-containing media was then harvested after a given number of days determined by strain (for both DENV3-CH53489 and ZIKV = 4 days post infection), centrifuged at $12,000 \times g$ for 45 min to remove cell debris, and pre-cipitated overnight with 8% (w/v) polyethylene glycol 8000 in NTE buffer (10 mM Tris-HCl, pH 8.0, 120 mM NaCl, 1 mM EDTA). Virus was then pelleted by centrifugation at $14,000 \times g$, the pellet resuspended in NTE buffer, then purified through a 24% sucrose cushion, and a final purification performed by ultracentrifugation using a 10–30% (w/v) potassium tartrate gradient. The virus band was extracted from the gradient using a syringe and buffer-exchanged into appropriate storage buffer (NTE) and concentrated using an Amicon Ultra-4 100 kDa centrifugal concentrator (Millipore). Preparation of immature DENV3-CH53489 was performed in the same manner but using Minimal Essential Media supplemented with 40 mM $NH_4Cl$ instead of RPMI-1640. Virus concentration and purity was estimated by running an SDS-PAGE gel and staining with Coomassie-blue with bovine serum albumin as a standard. All purification steps were performed at 4 °C.

**Quantification of the percentage of clubSP**. The population of normal virus particles and clubSP were manually counted in micrographs for DENV3-CH53489 incubated at 4, 10, 15, 20, 25, 30, 37 and 40 °C for 30 min. The percentage clubSP was determined as the proportion of clubSP from the total number of virus particles. The experiment was repeated three times. For each experiment, we counted at least nine micrographs at for each incubation temperature.

**Viral RNA protection assay**. Supernatant containing DENV3-CH53489 at 4 °C was transferred to different temperatures (4, 25, or 37 °C) and incubate for 30 min to induce clubSP formation in the 25 and 37 °C samples. RNase was then added to digest any RNA that has being extruded out into the supernatant. RNase activity was inhibited by addition of Buffer AVL from the QIAamp Viral RNA Kit before the virus was lysed and RNA extracted with the kit. Control conditions in which no RNase was added were also included. The viral RNA was then quantified through a one-step qRT-PCR using BioRad iTaq Universal SYBR Green One-Step kit and primers specific for DENV3-CH53489. The level of RNA in the sample is indicated by the quantification cycle (Cq) values of the experimental conditions subtracted from the Cq value of a control (no viral RNA) condition. The experiment was repeated twice with at least a triplicate in each independent experiment.

**Virus attachment assay**. C6/36 or BHK21 cells were seeded into wells of a 24-well plate at a density of $1 \times 10^6$ and $2 \times 10^5$ cells/well, respectively, and were allow to grow overnight. DENV3-CH53489, which was purified to cryoEM quality, was diluted by a factor of 1:1000 in cold RPMI containing 2% FBS and then incubated at 4 or 29 or 37 °C for 30 min (when incubated at 29 and 37 °C, virus will turn into clubSPs). Next, it was further incubated on ice for an additional 30 min. The inoculum was subsequently added at 1 MOG (multiplicity of genome copy number) to pre-cooled cells and incubate for an hour at 4 °C. The cells were then washed twice with ice-cold 1× PBS to remove unbound virus particles before subjected to RNA extraction using Qiagen RNAeasy Kit. The extracted RNA was

quantified via qRT-PCR using BioRad iTaq Universal One-Step RT-qPCR Kit and primers specific for DENV3-CH53489, and housekeeping genes, beta-actin and GAPDH for C6/36 and BHK cells, respectively. Negative control using uninfected cells were also included.

**Plaque reduction neutralization test (PRNT)**. The neutralization activity of the antibody 4G2 on DENV3-CH53489 clubSPs was determined by PRNT. DENV3-CH53489, purified to cryoEM quality, was incubated at 29 or 37 °C for 30 min to induce clubSPs. It was then incubated on ice for an additional 30 min. Next, fourfold serially diluted antibody 4G2 starting at 100 μg/ml was incubated with equal volumes of virus at 4 °C for 30 min. Hundred microliters of each mixture was then layered on pre-cooled BHK21 cells in a 24-well plate and incubated at 4 °C for an hour. The infected cells were washed with ice-cold 1× PBS before being overlaid with carboxyl-methyl cellulose and incubated at 37 °C for 4 days. Cells were fixed and stained, and the plaques were counted. Percentage neutralization was determined from the comparison of the number of plaques in specific antibody dilutions to the control (without antibody). $PRNT_{50}$ is the concentration of the antibody that causes 50% reduction in plaque numbers.

**CryoEM sample preparation**. For temperature-induced structural changes, virus was incubated at the given temperature for 30 min and then transferred to ice before application to a cryoEM grids. For generation of Fab-virus complexes, Fab was mixed with virus at a molar ratio of 1.5 Fab to E protein. The mixture was then incubated at the given temperature for 30 min and placed on ice before application to a cryoEM grid. Two microliters of sample was applied to copper cryoEM grids (with lacey carbon covered with thin carbon film), blotted for 3 s on filter paper, and plunged into a liquid ethane bath using a Vitrobot Mark IV plunger (FEI). The grids were then stored in liquid nitrogen or directly used.

**Data collection, image processing, and 3D reconstruction**. Images of vitrified virus or virus/Fab complexes were taken with a Titan Krios (FEI) electron microscope equipped with a 300 kV field emission gun at nominal magnifications of 47,000 for DENV and 59,000 for ZIKV. A $4096 \times 4096$ Falcon II (FEI) direct electron detector was used to record the images. Leginon was used for automated data collection[19]. Images for Fab C10:ZIKV were collected in movie mode with a total exposure of 1.6 s and total dose of 38 e⁻ Å⁻² , and the frames were aligned using MotionCorr[20]. Images of ~18 e⁻ Å⁻² total dose in the first frames were used in 3D reconstruction.

CTF-correction was performed with Gctf[21], and micrographs with poor defocus or astigmatism were discarded. For asymmetric reconstructions, particles were boxed in the EMAN2 program e2boxer[22]. Particles were then subjected to reference-free 2D-class averaging, 3D classification, and autorefinement in Relion-2[23]. For helical reconstruction, sliding window boxing of the segments was done by using the EMAN2 program e2helixboxer[22]. CryoEM helical reconstructions were done by using Relion-2[23,24]. Initial helical parameters were manually obtained by applying different transformation matrices in the program Chimera[25] under the 'sym' function, to place the Fab C10:E protein dimer model[13] (PDB 5H37) into the adjacent asu that best fit the map density. Three-dimensional classification was then performed with these initial parameters, allowing the values to vary within specified ranges. The resulting classes of maps were then examined for the clearest densities corresponding to the Fab. The parameters that generated the highest resolution density map were then chosen as the starting values for a further refinement restricting the helical parameters search within a narrower range than previously. The final map was obtained after the values had converged.

**Model building and refinement**. For Fab C10:ZIKV, a dimer of a previously published structure[13] (PDB 5H37) was able to be used directly after manually placing into the cryoEM density and fit using the command 'Fit in Map' in UCSF Chimera[25].

A homology model of DENV3-CH53489 E protein dimer was generated in SWISS-Model[26] based on the DENV3 cryoEM structure[27] (PDB 1UZG). This model and also the E protein dimer of the previously solved Fab C10:ZIKV were superimposed in Chimera and the coordinates of the Fab C10 was then combined with that of the DENV3-CH53489 to generate the DENV3-Fab C10 model. This combined model was then manually fitted into the cryoEM density of the Fab C10:DENV3-CH53489 clubSP and the fit was optimized by using the command 'Fit in Map' in Chimera.

Inter-dimer contacts were mapped by measuring the distance between Cα residues using a cutoff distance of 8.0 Å in CCP4[28]. Superposition of structures were performed in Chimera using the MatchMaker function.

**Reporting summary**. Further information on research design is available in the Nature Research Reporting Summary linked to this article.

## Data availability

The cryoEM maps and coordinates have been deposited in the Protein Data Bank (PDB) and Electron Microscopy Data Bank (EDMB); DENV3-CH53489 with Fab C10 (EMDB: 30278, PDB: 7C2S) and ZIKV with Fab C10 (EDMB: 30279, PDB: 7C2T). Data underlying Figs. 1b, 2a, b, Supplementary Figs. 4a and 5 are provided in the Source Data

file. All other data are available from the corresponding author upon reasonable request. Source data are provided with this paper.

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

## Acknowledgements

We thank Aravinda de Silva for providing 8A1 antibody and Guntur Fibriansah for technical support. This work is supported by National Research Foundation Investigatorship award (NRF-NRFI2016-01) and National Research Foundation competitive Research Project grant (NRF2016NRF-CRP001-063) awarded to S.-M.L. and the Duke-NUS Signature Research Programme funded by the Ministry of Health, Singapore. This work was supported by Public Health Service grants GM122979 and GM127365 awarded to M.C.M. from the National Institutes of Health, USA.

## Author contributions

S.-M.L. and M.C.M. co-supervised the project. G.S. provided antibodies. V.S.Y.C., X.-N.L., S.R.M., J.L., J.L.T., J.W. and P.-L.C. purified virus. M.W. performed the RNA protection assays. V.S.Y.C performed the attachment assays. S.R.M., P.-L.C., S.Z. and T.Y.T. purified Fab and complexed virus with Fab. S.-M.L. counted virus populations. S.R.M. and T.-S.N. froze samples for cryoEM. T.-S.N. and J.S. collected cryoEM data. S.R.M. performed the cryoEM image reconstructions. S.R.M., V.A.K., M.C.M., and S.-M.L. interpreted the maps. S.R.M., S.-M.L., and M.C.M. wrote the manuscript. All authors gave feedback on the manuscript.

## Competing interests

The authors declare no competing interests.
