## [Peer Review File · Nature Communications]

Reviewers' comments:

Reviewer #1 (Remarks to the Author):

This manuscript provides EM structural analysis of a different morphology of flaviviruses, a club-shaped particle, which exists at mosquito and human body temperature. Most prior structural studies of flaviviruses have focused on samples held at 4 degrees which do not exhibit these structures. Notably, the club-shaped article accounts for half the DENV3 virions at relevant body temperature.

The club-shaped morphology likely presents antibody epitopes differently which will affect antibody efficacy. Prior studies in the field have demonstrated for example, neutralizing antibodies against quaternary epitopes and enhancing antibodies against other sites involving the fusion loop. Both could be affected by the E protein arrangement in the mosquito and human body-temperature morphology. For example, the fusion loop is hidden in the 4 degree structure but appears exposed in the body temperature structure. Fab 8A1 binds only to the head and tail, meaning its epitope is in a different structure in the club-shaped body. In contrast, Fab C10 binds throughout the particle (also indicating that the novel morphology is formed of E like the 4 degree spherical morphology).

Provision and characterization of this new model is an important advance for the flavivirus community and examines a new-to-us but physiologically relevant form of the virion that one may see in the mosquito vector and the human host. Antibody binding studies demonstrate that these particles are adopted by mature, rather than immature viruses. Other dengue viruses examined also show the presence of club-shaped particles as well suggesting this is a broadly applicable phenomenon at temperature relevant for infection, although in this study, the DENV3 has a greater propensity for club-shaped particle formation than other types of DENV. High fever temperature (40 degrees) seems to produce more club-shaped articles from DENV2. Interestingly, the structure adopted is irreversible (does not revert to spherical if temp. is dropped). Both club-shaped and spherical viruses attach to cells equally well.

EM structures were determined for a club-shaped particle of DENV3 and one from ZIKV and found to have different rotations (24 vs 102 degrees) and different axial rise. This leads to a different arrangement of protomers of E relative to each other in the rafts. Do the DENV3 particles always have thinner shaft while ZIKV broader?

These results here are a significant advance for the flavivirus field as they demonstrate an alternate arrangement of E at relevant vector, healthy body and feverish body temperatures that must be considered when interpreting antibody elicitation and antibody reactivity, as well as vaccine particle production. This paper provides a key new model and a key new direction for the field.

The writing and figure presentation could be a bit clearer as it was a bit difficult to wade through. Editing would better highlight the advance provided by this study.

Line 83, add "in this morphology" to read "we solved cryoEM helical structures of an antibody-virus complex for both DENV and ZIKV in this morphology, and reveal the architecture..."

Line 95, Why does high osmolarity preferentially lyse club-shaped particles?

Line 115, Are CH53489 and 8A1 neutralizing?

Line 166, sentence split in two

Figure 1c, add labels to left and right panels of 4° and 37°

Figure 1d, add labels of DENV and ZIKV

Figure 1e, label head and tail- are these class averages or the individual particle in the red box?

Figure 2, enlarge +/- RNase key

I'm confused the experiment in Figure 2a. Were the viruses transferred from 4 to the indicated temps, then lysed vs lysed at 4 then transferred to the indicated temperatures? Please clarify the Figure legend and label the experimental panels better.

Figure 3, label ZIKV on panels a-c and DENV3 on d-e

Figure 4 is fascinating. Would it be possible to add another panel with circles outlining the epitopes of one or known quaternary neutralizing and enhancing antibodies?

EO Sapphire

Reviewer #2 (Remarks to the Author):

The manuscript by Morrone et.al., describes a new morphological variant of Dengue and Zika virus which is markedly different from the accepted icosahedral structure of these viruses. The paper identifies and characterizes the possible structure of this morphology using Fabs against the E protein. The manuscript overall reports on an interesting morphology of flaviviruses that would have an impact on how we envision the virus infectivity and immune response, but needs more explanations. Some comments and suggestions for the authors are below:

1. How old were the virus samples after purification when conducting these experiments?

Flaviviruses and other membrane enveloped viruses have been known to show changes in their morphology and heterogeneity with time after purification. Figure 1a showing the virus prep at 4C already shows many broken and damaged particles, prompting the question as to the quality of the virus prep during experiments.

2. Line 93-96: The authors suggest that the absence of clubSPs on purification from mosquito cells at 29C could be due to the clubSPs being lysed in the high osmolarity of purification steps. But they have no evidence to support this claim. There could be other cellular factors or serum components during virus preparation that prevents the virus from forming clubSPs. This statement should be modified to include other possibilities or removed as it implies that the clubSPs are more fragile than regular virus particles, for which there is no evidence provided.

3. Line 100-101: How was the percentage of population of clubSPs calculated? Was it calculated by counting particles from cryo-EM micrographs? If yes, then what was the sample size from which the numbers were calculated? These need to be explained in the methods to prove the significance of the values.

4. Line 101: The text reads Supplemental fig 1b whereas the actual data is part of Main figure 1b.

5. DENV2-PVP94/07 strain does not show much clubSP formation, but rather obvious breakdown and clumping at higher temperatures. DENV1-WestPac/s clubSPs have markedly different elongated structure than that of DENV-3. What happens when the authors incubate the virus for longer than 30 minutes at temperatures above 29C? Could this tubular morphology of virus particles be a structural stage before the virus loses structural integrity at higher temperatures? For example, does the virus go from smooth to bumpy and from there to this tubular structure?

6. In Fig.1c showing binding of Fab 8A1 to DENV post incubation at 37C, the increased spikiness of the head and tail of the clubSPs presumably due to Fab binding is hard to see in the image. Addition of a zoomed-in inset comparing it with the uncomplexed control would highlight the point effectively. In the current images, it is hard to accept the claim.

7. Line 124: In continuation to the above comment, the authors claim that the DIII domain of E protein is more exposed in the head and tip of tail of ClubSPs, which allows Fab 8A1 to bind. But how are they sure that all parts of the extended membraneous structure is covered with protein?

What if they are uncovered patches of exposed membrane in these tubular virions? How do the authors account for it? Even in the case of Fab C10, where the Fab binding throughout the clubSP particles is obvious, the images have been acquired by heating the virus post Fab binding, which would have influenced the arrangement of the E proteins as discussed in the paper. Do the authors have any evidence that the uncomplexed, native clubSPs have a continuous protein cover? It is hard to imagine it given the varied dimensions of the clubSPs.

8. Line 127-130: The authors claim that antibody 4G2 binds the clubSPs based on increased spikiness of the particles in the micrographs, but again this is very subjective. At the current zoom of the images, it is borderline to claim that 4G2 binds the clubSPs. At the minimum, zoomed insets comparing the uncomplexed and complexed particles are needed.

9. Why treat head of DENV as cylinder and not sphere? Is the cylindrical nature of the head obvious? Or is it just to minimize the volume calculations?

10. Line 300-302 in Discussion: The authors state in the manuscript that the Fab was added to the virus and then the complex heated for 30min at higher temperatures to form the Fab bound clubSP and catSP structures, which were then used for reconstruction. In this scenario, the statement in line 300-302 cannot be stated or is irrelevant as when the Fab binds to the virus in the experiment, the virus is in its spherical, icosahedral form. Thus, the E protein are in the dimeric raft formation already.

Reviewers' comments:

Reviewer #1 (Remarks to the Author):

This manuscript provides EM structural analysis of a different morphology of flaviviruses, a club-shaped particle, which exists at mosquito and human body temperature. Most prior structural studies of flaviviruses have focused on samples held at 4 degrees which do not exhibit these structures. Notably, the club-shaped article accounts for half the DENV3 virions at relevant body temperature.

The club-shaped morphology likely presents antibody epitopes differently which will affect antibody efficacy. Prior studies in the field have demonstrated for example, neutralizing antibodies against quaternary epitopes and enhancing antibodies against other sites involving the fusion loop. Both could be affected by the E protein arrangement in the mosquito and human body-temperature morphology. For example, the fusion loop is hidden in the 4 degree structure but appears exposed in the body temperature structure. Fab 8A1 binds only to the head and tail, meaning its epitope is in a different structure in the club-shaped body. In contrast, Fab C10 binds throughout the particle (also indicating that the novel morphology is formed of E like the 4 degree spherical morphology).

Provision and characterization of this new model is an important advance for the flavivirus community and examines a new-to-us but physiologically relevant form of the virion that one may see in the mosquito vector and the human host. Antibody binding studies demonstrate that these particles are adopted by mature, rather than immature viruses. Other dengue viruses examined also show the presence of club-shaped particles as well suggesting this is a broadly applicable phenomenon at temperature relevant for infection, although in this study, the DENV3 has a greater propensity for club-shaped particle formation than other types of DENV. High fever temperature (40 degrees) seems to produce more club-shaped articles from DENV2. Interestingly, the structure adopted is irreversible (does not revert to spherical if temp. is dropped). Both club-shaped and spherical viruses attach to cells equally well.

EM structures were determined for a club-shaped particle of DENV3 and one from ZIKV and found to have different rotations (24 vs 102 degrees) and different axial rise. This leads to a different arrangement of protomers of E relative to each other in the rafts. Do the DENV3 particles always have thinner shaft while ZIKV broader?

The ZIKV:C10 complex has a catSP morphology instead of the clubSP. Because the uncomplex ZIKV particle looks similar to the DENV3 clubSP, and our previous work shows that C10 binds across E proteins at both the intra- and inter-dimer interfaces, we think the ZIKV:C10 catSP morphology is a result of C10 locking all E proteins within a raft together. To answer reviewer #1 question whether uncomplexed DENV3 and uncomplexed ZIKV shaft has the same dimension, we measure the tail of them, and the width of the DENV3 is ~160Å and the ZIKV is ~150Å.

We have now included a line (line number 155-156):

“We measured the width of the tail of the uncomplexed ZIKV clubSP (~150Å) and they are similar to the DENV3 clubSP (~160Å).”

These results here are a significant advance for the flavivirus field as they demonstrate an alternate arrangement of E at relevant vector, healthy body and feverish body temperatures that must be considered when interpreting antibody elicitation and antibody reactivity, as well as vaccine particle production. This paper provides a key new model and a key new direction for the field.

We thank reviewer #1 for emphasizing on its importance.

The writing and figure presentation could be a bit clearer as it was a bit difficult to wade through. Editing would better highlight the advance provided by this study.

Line 83, add “in this morphology” to read “we solved cryoEM helical structures of an antibody-virus complex for both DENV and ZIKV in this morphology, and reveal the architecture...”

Corrected to

“We solve the cryoEM helical structures of an antibody-virus complex for both DENV and ZIKV in this morphology and reveal the architecture of these structures.”

Line 95, Why does high osmolarity preferentially lyse club-shaped particles?

The purified virus after incubation at 29°C consistently showed increased amounts of ClubSP. Therefore we inferred that since the virus was originally cultured in mosquito cell lines at 29°C prior to purification, the presence of the low amount of clubSP after purification, could be due to the instability of clubSP particles in the high osmolarity conditions during the purification process. Recently, we have done more virus purification but with a different personnel working on it, for one of these preps, we did obtain more clubSP at 4°C, it suggests perhaps how gentle the person is handling the virus could make a difference. However, as reviewer #1 and #2 pointed out, we do not have strong evidence to understand why some preps immediately after purification (the 4°C control) have more or less clubSP. Regardless, all purified virus preps, in the different subsequent incubation experiments are consistent with each other – there are increased in the number of clubSP particles at elevated temperatures.

As reviewer #1 and #2 pointed out, since we do not know the real reasons for why we see much less ClubSP in the virus immediately right after purification, we have deleted this sentence from the manuscript.

Line 115, Are CH53489 and 8A1 neutralizing?

We have carried out a neutralization assay of 8A1 against DENV3 CH53489 and have included it in our result section.

“One of the antibodies is a neutralizing mouse monoclonal antibody 8A1¹⁵, we have determined its PRNT₅₀ against DENV3-CH53489 to be 0.11µg/ml.”

Line 166, sentence split in two

Sentence changed to:

“It is possible that the formation of DENV3-CH53489 clubSPs is due to the loss of its RNA genome. We detect for the release of RNA outside the viral particle after induction of clubSP, by their sensitivity towards added RNase – viral genome RNA inside particle are protected from RNase while those released are not. We first incubated the virus at different temperatures and then conducted RNase digestion.”

Figure 1c, add labels to left and right panels of 4° and 37°.

Corrected

Figure 1d, add labels of DENV and ZIKV

Corrected

Figure 1e, label head and tail- are these class averages or the individual particle in the red box?

Yes, they are class averages. We modified the legends:

“e, Micrograph showing the DENV3-CH53489 clubSPs (red box) are mostly mature virus, as anti-prM Fab DV62.5 binding was not detected in the 2D averages (bottom panels) of either the head or tail of the clubSP particle. Fab DV62.5 was observed to bind to the spherical spiky immature virus population (black box) as shown in Supplementary Fig. 2c.”

Below is the new Figure 1.

Figure 2, enlarge +/- RNase key

We have corrected that. Please see below new figure 2.

I'm confused the experiment in Figure 2a. Were the viruses transferred from 4 to the

indicated temps, then lysed vs lysed at 4 then transferred to the indicated temperatures?
Please clarify the Figure legend and label the experimental panels better.

We have edited the legends to make it clearer:

“Fig. 2 DENV3-CH53489 clubSPs contain RNA and remain infectious.

a, The viral DENV3-CH53489 RNA genome is not extruded out of the particle after temperature-induced structural change at 25 °C and 37 °C. (Left panel) Viruses at 4°C were transferred to different temperatures and incubated for 30 mins. RNase is then added to digest away any RNA genome that was extruded outside the virus. RNase activity is inhibited and then the virus was lysed. Detection of a 260-nucleotide stretch of RNA genome using a primer set in qRT-PCR indicates intact viral genome. In all temperatures, the amount of genome with and without addition of RNase appeared the same, suggesting that viral RNA remains within the particle after temperature treatment and were thus protected from RNase treatment. The y-axis is expressed as difference in Cq values from virus-free reference. (Right panel) Control experiments to show when virus is lysed first before exposure to RNase, the RNases can successfully break down viral RNA genome. Results showed when RNA genome (due to lysis) are exposed, RNases digestion occurs and there are much lower amounts of intact RNA genome detected by qRT-PCR than when no RNase is added. Standard deviations are from two independent experiments.”

We also included the design of experimental procedure above each panel to make it clearer. Below is the new Figure 2.

Figure 3, label ZIKV on panels a-c and DENV3 on d-e

This is corrected.
See below.

Below is the new Figure 3.

Figure 4 is fascinating. Would it be possible to add another panel with circles outlining the epitopes of one or known quaternary neutralizing and enhancing antibodies?

We have added a new figure in Supplementary Figure 1e- see below figure. We have circled one epitope bound by C10 (black), 2A10G6 (fusion loop), and 5J7 (pink) on the E proteins inside a raft.

We have added to the legend for Supplementary Figure 1e.

“(e) Epitopes bound by antibodies C10, 2A10G6 (fusion loop), and 5J7 on the E proteins inside a raft are circled and shaded in black, green and pink, respectively.”

We have also added discussion on this:

“Results presented by Rey and colleagues¹⁷ showed that when C10 is added to mostly monomeric soluble E proteins, the C10 Fabs can assemble the E proteins into dimers. Whether the E protein protomer interactions within a dimer in the uncomplexed DENV3:clubSP are also weak is hard to determine, but the protomers should be near each other so that Fab C10 can assemble the dimer. Other than C10, there is also another reported quaternary structure dependent binding human monoclonal antibody 5J7¹² which binds across three E proteins on the spherical compact virus surface (Supplementary Figure. 1e). This epitope would be completely disrupted in the tail part of the DENV3 clubSP suggesting 5J7 is likely unable to bind. An enhancing mouse monoclonal antibody 2A10G6 binds to the fusion loop of the E protein (Supplementary Figure. 1e), as the E proteins are more loosely packed in the tail part of clubSP, it may be able to bind more efficiently.”

EO Sapphire

Reviewer #2 (Remarks to the Author):

The manuscript by Morrone et.al., describes a new morphological variant of Dengue and Zika virus which is markedly different from the accepted icosahedral structure of these viruses. The paper identifies and characterizes the possible structure of this morphology using Fabs against the E protein. The manuscript overall reports on an interesting morphology of flaviviruses that would have an impact on how we envision the virus infectivity and immune response, but needs more explanations. Some comments and suggestions for the authors are below:

1. How old were the virus samples after purification when conducting these experiments? Flaviviruses and other membrane enveloped viruses have been known to show changes in their morphology and heterogeneity with time after purification. Figure 1a showing the virus prep at 4C already shows many broken and damaged particles, prompting the question as to the quality of the virus prep during experiments.

We used the exactly the same purification procedure as that for purifying DENV2 with smooth (DENV2 PVP94/07) or bumpy surface (DENV2 NGC) morphologies. Also similar to the other preps, the virus particles are imaged < 1 day after purification. We have repeated the experiments over many years and the results are consistent and reproducible (see below figures). Maybe this particular prep has more damaged particles at 4°C, however, from the figures below, we can see the other DENV3 CH53489 preps that have less broken particles and are still showing clubheads.

We have changed Figure 1a to a micrograph that shows a cleaner sample but it still reacts the same way, in terms of clubshape particle formation upon incubation at higher temperatures. (see Figure 1 in the reply to reviewer #1)

Below we show three different preparations for each of the DENV2 smooth (PVP94/07) and bumpy surface (NGC) morphology viruses and also for DENV3 CH53489.

2. Line 93-96: The authors suggest that the absence of clubSPs on purification from

mosquito cells at 29C could be due to the clubSPs being lysed in the high osmolarity of purification steps. But they have no evidence to support this claim. There could be other cellular factors or serum components during virus preparation that prevents the virus from forming clubSPs. This statement should be modified to include other possibilities or removed as it implies that the clubSPs are more fragile than regular virus particles, for which there is no evidence provided.

Thank you for pointing this out, reviewer #1 also has the same concern. We have now removed this sentence from the manuscript- also see reply to reviewer #1.

3. Line 100-101: How was the percentage of population of clubSPs calculated? Was it calculated by counting particles from cryo-EM micrographs? If yes, then what was the sample size from which the numbers were calculated? These need to be explained in the methods to prove the significance of the values.

The old figure was from one experiment. We now repeated the experiment three times and have made a new Figure 1b (see figure in reply to reviewer #1 section), now it includes standard deviation.

The population of clubSP were manually counted from at least 9 micrographs (about 100-250 individual particles per micrograph) collected for each temperature (4, 10, 15, 20, 25, 30, 37, 40).

The conclusion has changed as when we repeated the new results suggest a gradual increase in % of clubSP with temperature and the maximum percentage is reached at 37°C.

We changed the legend of Figure. 1 b to "Graph showing percentage of clubSPs increase gradually with temperature. The maximum percentage of clubSP is reached at 37°C. Standard deviations were calculated from three individual experiments."

We have changed the result section (line 102) to:

"We observed gradual increase in percentage of clubSP virus population, the maximum is reached at 37°C (Fig. 1b)."

We also included a section in the methods:

"Quantification of the percentage of clubSP at a different temperatures

The population of normal virus particles and clubSP were manually counted in micrographs for DENV3-CH53489 incubated at 4, 10, 15, 20, 25, 30, 37 and 40 °C for 30 mins. Percentage clubSP was determined as the proportion of clubSP from the total number of virus particles. The experiment was repeated as three times. For each experiment, we counted at least 9 micrographs at for each incubation temperature."

4. Line 101: The text reads Supplemental fig 1b whereas the actual data is part of Main figure 1b.

Thanks for spotting the mistake. We have now corrected it.

5. DENV2-PVP94/07 strain does not show much clubSP formation, but rather obvious breakdown and clumping at higher temperatures. DENV1-WestPac/s clubSPs have markedly different elongated structure than that of DENV-3.

What happens when the authors incubate the virus for longer than 30 minutes at temperatures above 29C? Could this tubular morphology of virus particles be a structural stage before the virus loses structural integrity at higher temperatures? For example, does the virus go from smooth to bumpy and from there to this tubular structure?

We conducted an experiment to incubate DENV3-CH53489 for different length of time at 37°C (previous incubation time for 30 mins). At all incubation time (15mins, 30mins, 1hr, 1.5hr, 2 hrs), the fraction of virus with clubSPs appears to be similar. We therefore do not think that the virus transit through the smooth to bumpy surface morphology first before turning into clubSP. Even after 2 hrs, the virus is still not disrupted and the fraction of the clubSP is the same.

We have added in the result to describe this, under the section “DENV exhibits temperature-dependent structural changes”:

“We also tried different incubation time (15 mins to 2hrs) at 37°C to determine if more or less clubSP will form. Results showed that the fraction of particles turning to clubSP are the same regardless of incubation time (Supplementary Fig. 2c).”

Below shows the new figure in Supplementary figure 2c.

6. In Fig.1c showing binding of Fab 8A1 to DENV post incubation at 37C, the increased spikiness of the head and tail of the clubSPs presumably due to Fab binding is hard to see in the image. Addition of a zoomed-in inset comparing it with the uncomplexed control would highlight the point effectively. In the current images, it is hard to accept the claim.

We have enlarged the image and also filtered it to improve the contrast. The binding of Fab to the head and the tip of the tail of clubSP is much clearer. – see updated Figure 1c in the reply to reviewer #1.

7. Line 124: In continuation to the above comment, the authors claim that the DIII domain of E protein is more exposed in the head and tip of tail of ClubSPs, which allows Fab 8A1 to bind. But how are they sure that all parts of the extended membraneous structure is covered with protein? What if they are uncovered patches of exposed membrane in these tubular virions? How do the authors account for it? Even in the case of Fab C10, where the Fab binding throughout the clubSP particles is obvious, the images have been acquired by heating the virus post Fab binding, which would have influenced the arrangement of the E proteins as discussed in the paper. Do the authors have any evidence that the uncomplexed, native clubSPs have a continuous protein cover? It is hard to imagine it given the varied dimensions of the clubSPs.

We conducted experiments whereby we first induced clubSP by heating the DENV3 to 37°C and then we add Fab C10 to it. We observed C10 to bind throughout the virus particle similar to that observed when the virus is first mixed with C10 then incubated at 37°C. This indicates that the E protein are present throughout the clubSP surface.

We have added a line in the result section, under the section “Antibodies recognize the flavivirus structural variant”.

“When we added Fab C10 to DENV3-CH53489 and then increased the temperature to 37 °C, half of the virus particles become clubSPs, similar to the uncomplexed controls, except Fab was observed to bind to the entire surface of the clubSPs (Fig. 1d, left). The same is also observed when the virus is incubated first at 37°C to induced clubSP and then Fab C10 is added (Supplementary Fig. 4c).”

Below is the Supplementary Fig. 4c.

8. Line127-130: The authors claim that antibody 4G2 binds the clubSPs based on increased spikiness of the particles in the micrographs, but again this is very subjective. At the current zoom of the images, it is borderline to claim that 4G2 binds the clubSPs. At the minimum, zoomed insets comparing the uncomplexed and complexed particles are needed.

We have enlarged the image and also filtered to increase contrast- see Supplementary Fig. 4a and figure below.

a

DENV3-CH53489 control

DENV3-CH53489 + Fab4G2

9. Why treat head of DENV as cylinder and not sphere? Is the cylindrical nature of the head obvious? Or is it just to minimize the volume calculations?

We used cylinder for calculations because the whole particle C1 reconstruction shows side-view of the head is flat (below figure left). We can also observe from the projections (below figure right) that the head is flat: blue box shows side-view, red box showed top view.

Side-view

top-view

We now clarify this by including a sentence in the first paragraph of the discussion.

“Measurement of the head of the 2D average of the DENV3 clubSP (Supplementary Fig. 2a) and calculation of its volume treating it as a cylinder suggests that there is room to accommodate about two copies of viral genome (Supplementary Table 1). The cylindrical shape of the clubSP head was observed in our asymmetric reconstructed map (Fig. 1f).”

10. Line 300-302 in Discussion: The authors state in the manuscript that the Fab was added to the virus and then the complex heated for 30min at higher temperatures to form the Fab bound clubSP and catSP structures, which were then used for reconstruction. In this scenario, the statement in line 300-302 cannot be stated or is irrelevant as when the Fab binds to the virus in the experiment, the virus is in its spherical, icosahedral form. Thus, the E protein are in the dimeric raft formation already.

Reviewer #2 is referring to the below sentences:

“Results presented by Rey and colleagues¹⁷ showed that when C10 is added to mostly monomeric soluble E proteins, the C10 Fabs can assemble the E proteins into dimers. Whether the E protein protomer interactions within a dimer in the uncomplexed DENV3:clubSP are also weak is hard to determine, but the protomers should be near each other so that Fab C10 can assemble the dimer.”

We thank reviewer #2 for pointing this out. Now we have conducted the experiment whereby we first induced ClubSP particles before adding C10 Fab and it shows that C10 could still bind to the tail of the ClubSP particles and hence this conclusion is still valid. Also see reply to reviewer #2, question (7).

REVIEWERS' COMMENTS:

Reviewer #1 (Remarks to the Author):

All my comments have been addressed.

Reviewer #2 (Remarks to the Author):

The revised manuscript by Morrone et.al., answers my previous queries satisfactorily. I only have a minor comment: Figure 1 has numerous panels referring to different experimental conditions and referring to them as 'right' and 'rightmost' makes it a bit hard to follow. It would be better if the authors could label each panel and refer to them distinctly.

We labelled them a (i), (ii) and (iii) etc. to distinguish them and corrected the figure reference in the main text.